# A New Laboratory Workflow Integrating the Free Light Chains Kappa Quotient into Routine CSF Analysis

**DOI:** 10.3390/biom12111690

**Published:** 2022-11-15

**Authors:** Malte Johannes Hannich, Mohammed R. Abdullah, Kathrin Budde, Astrid Petersmann, Matthias Nauck, Alexander Dressel, Marie Süße

**Affiliations:** 1Institute of Clinical Chemistry and Laboratory Medicine, University Medicine Greifswald, 17475 Greifswald, Germany; 2Friedrich Loeffler-Institute of Medical Microbiology, University Medicine Greifswald, 17475 Greifswald, Germany; 3Institute for Clinical Chemistry and Laboratory Medicine, University Oldenburg, 26133 Oldenburg, Germany; 4Department of Neurology, Carl-Thiem Klinikum Cottbus, 03048 Cottbus, Germany; 5Department of Neurology, University Medicine Greifswald, 17475 Greifswald, Germany

**Keywords:** cerebrospinal fluid, free light chain kappa, immunoglobulin, laboratory workflow, protein analytic

## Abstract

We performed this cohort study to test whether further analysis of intrathecal inflammation can be omitted if the free light chain kappa (FLCκ) quotient is within the reference range in the corresponding quotient diagram. FLCκ concentrations were measured in serum and cerebrospinal fluid (CSF) samples. The intrathecal fraction (IF) of FLCκ was calculated in relation to the hyperbolic reference range. 679 patient samples were used as a discovery cohort (DC). The sensitivity and negative predictive value (NPV) of the FLCκ-IF for the detection of an intrathecal humoral immune response (CSF-specific OCB and/or IF IgG/A/M > 0%) was determined. Based on these data, a diagnostic algorithm was developed and prospectively validated in an independent validation cohort (VC, *n* = 278). The sensitivity of the FLCκ-IF was 98% in the DC and 97% in the VC with a corresponding NPV of 99%. The use of the FLCκ-IF as a first line analysis would have reduced the Ig and OCB analysis by 62% in the DC and 74% in the VC. The absence of a FLCκ-IF predicts the absence of a humoral intrathecal immune response with a very high NPV of 99%. Thus, integration of our proposed algorithm into routine CSF laboratory analysis could help to reduce analytical efforts.

## 1. Introduction

The detection of an intrathecal humoral immune response is an important diagnostic aspect in many neurological diseases of the central nervous system (CNS). One method to represent this is the quantitative protein analysis of immunoglobulins. It is usually performed using quotient diagrams in which the cerebrospinal fluid (CSF) and serum quotient of immunoglobulins (IgG, IgA, IgM) is related to the CSF and serum albumin quotient. These calculations are necessary to distinguish the fractions of proteins that enter the CSF from blood by passive diffusion from those produced intrathecally [1]. Another approach is the qualitative analysis of IgG by isoelectric focusing and immunoblotting, the determination of the presence of oligoclonal bands (OCB) in CSF [2]. The free light chains kappa (FLCκ) in CSF as biomarker for intrathecal inflammation has recently gained renewed interest. At least in Multiple Sclerosis (MS), measurement of FLCκ is superior to quantitative Ig analysis in terms of sensitivity for detection of intrathecal inflammation. In comparison with OCB analysis, the measurement of FLCκ provides the advantage of comparable sensitivity, the possibility of a fully automated measurement, and objective as well as quantitative value interpretation rather than the qualitative and therefore subjective OCB interpretation [3,4,5,6,7,8,9]. Its diagnostic specificity though is clearly inferior. OCB provide information about intrathecal IgG synthesis, FLCκ synthesis can also be seen in an intrathecal IgA- and IgM-synthesis [10]. 

Despite a broad variety of publications about the use of FLCκ in CSF in the diagnostics of MS and other neurological diseases, integration into the diagnostic routine in CSF laboratories has not yet been undertaken. We hypothesized that in the absence of intrathecal FLCκ synthesis, a qualitative or quantitative Ig synthesis (hereafter referred to as an intrathecal humoral immune response) should also not be detectable. Therefore, the main objective of this work was to evaluate a laboratory workflow that integrates the presence of intrathecal FLCκ synthesis as a highly sensitive marker with a strong negative predictive value for an intrathecal humoral immune response. This has the potential to minimize the qualitative and quantitative determination of Ig in laboratory routine CSF diagnostics.

## 2. Materials and Methods

### 2.1. Standard Protocol Approvals and Patient Consent

The retrospective part of the study was based on the votum of the local ethics research committee (III UV 39/03 and BB019/18; University Medicine Greifswald, Greifswald, Germany). For the prospective part of the study, consecutive CSF and serum samples were collected between 3 January 2019 and 28 May 2019, based on the votum of the local ethics research committee (BB117/18; University Medicine Greifswald, Greifswald, Germany). Written informed consent was obtained from all participants. 

### 2.2. Patient Samples and Cohorts

All samples were acquired from patients of the Department of Neurology, University Medicine Greifswald (Greifswald, Germany). For this cohort study, a two-step study design was used: First, a discovery cohort (*n* = 679) was established to determine sensitivity and negative predictive value of the FLCκ-IF to predict an intrathecal humoral immune response. Based on these data, a laboratory workflow integrating the FLCκ as a first line analysis for detection of an intrathecal humoral immune response was created. The discovery cohort consisted of patient samples which had been acquired prospectively between 2015 and 2017, as well as samples collected between 2008 and 2016, which were stored at −80 °C. This novel workflow was then prospectively evaluated with an independent validation cohort (*n* = 278), which has been acquired consecutively between 3 January 2019 and 28 May 2019. 

Some patient samples of the discovery- and validation cohort have already been reported in the context of other studies with a different focus [5,10,11,12,13]. 

Patient samples of the discovery and validation cohort presenting signs of either CSF specific OCB and/or an IF of IgG, A, or M > 0% were considered to have an intrathecal humoral immune response. 

Blood admixture to the CSF can occur due to intrathecal haemorrhage or artificially due traumatic lumbar puncture. Such blood admixture can mimic intrathecal Ig synthesis in the absence of intrathecal inflammation. Ig synthesis due to blood admixture was diagnosed in patient samples presenting the following findings: Erythrocyte count (EC) > 500/µL and QIgM > Qlim (IgM) and if present QIgA > Qlim (IgA) and QIgG > Qlim (IgG) but IF IgM > IF IgA > IF IgG [13]. The resulting Ig synthesis in the corresponding quotient diagram was therefore not considered due to an intrathecal humoral immune response.

Characteristics of the discovery and validation cohort including CSF results are presented in Table 1.

### 2.3. Laboratory Analyses

Laboratory analyses were performed in the interdisciplinary CSF laboratory of the University Medicine Greifswald as described previously [12].

As a qualitative method to detect intrathecal IgG synthesis, the analysis of OCB was done by isoelectric focusing with a semiautomatic agarose electrophoresis system (Hydragel 9 CSF, Sebia Hydrasys 2Scan, Sebia GmbG, Fulda, Germany). 

FLCκ in sera and CSF were measured by nephelometry with the N Latex FLC kappa kit (Siemens Healthcare Diagnostics Products GmbH, Marburg, Germany), using monoclonal antibodies on the BN Prospec analyzer. CSF pre-dilution was set to 1:2, and serum pre-dilution was set to 1:100. The lower limit of quantification of the assay was 0.034 mg/L. 

One major concept in CSF diagnostics is to distinguish the serum fraction of proteins from intrathecally synthesized proteins (IgG, IgA, IgM or FLCκ). Therefore, the interpretation of Ig and FLCκ quotients in quotient diagrams is a well-established method [3,10]. In these quotient diagrams, immunoglobulin and FLCκ CSF and serum quotients (QIgG, QIgA, QIgM, QFLCκ; y-axis) are plotted in reference to the albumin CSF and serum quotient (QAlb; x-axis). 

The hyperbolic reference range, as well as the intrathecal fraction of Ig and FLCκ was calculated according to the formulas described by Reiber et al. [1,2,3]: Q_lim (IgG) = 0.93 × √((QAlb)^2^ + 6 × 10^−6^) − 1.7 × 10^−3^
Q_lim (IgA) = 0.77 × √((QAlb)^2^ + 23 × 10^−6^) − 3.1 × 10^−3^
Q_lim (IgM) = 0.67 × √((QAlb)^2^ + 120 × 10^−6^) − 7.1 × 10^−3^
Q_lim (FLCκ) = 3.27 × √((QAlb)^2^ + 33 × 10^−6^) − 8.2 × 10^−3^
Ig/FLCκ-IF = [1 − Qlim(Ig/ FLCκ)/QIg/FLCκ] × 100 [%].

The upper line (Qlim) of the reference range discriminates between the blood derived- and intratecally synthesized- Ig and FLCκ fraction.

### 2.4. Statistical Analysis

Data from the discovery cohort were used to establish an evaluation algorithm by estimating the sensitivity and the negative-predictive value of the presence of an intrathecal fraction of FLCκ for the prediction of an intrathecal humoral immune response as defined above. 

RStudio (R version 3.5.1 2 July 2018) was used for statistical and graphical processing of the data.

## 3. Results

### 3.1. Establishing a New Diagnostic Workflow Integrating FLCκ-IF as a First Line Analysis for an Intrathecal Humoral Immune Response

A total of 679 patient samples were identified to form the discovery cohort (Table 1, Figure 1). Signs of an intrathecal humoral immune response as defined above were detected in 245 of these samples. Of those, 241 had an intrathecal synthesis of FLCκ, accounting for 98% of the samples with an intrathecal humoral immune response. The remaining 4 (~2%) had no evidence of an intrathecal synthesis. Of these 4 patient samples, 3 presented CSF specific OCB and had a FLCκ serum value exceeding the reference range (according to manufacturer’s specification: 6.7–22.4 mg/L) (patient1: 57.7 mg/L, patient2: 66.9 mg/L, patient3: 57.1 mg/L). One patient sample though had a borderline IgM synthesis (IF IgM: 10%) that could not be explained by blood contamination or higher serum FLCκ values. 434 patient samples (64%) did not show signs of an intrathecal humoral immune response as defined above. Of these, 60 patient samples had an intrathecal synthesis of FLCκ (9%). 

The use of the IF FLCκ as a primary selection marker in this cohort would have resulted in a sensitivity of 98%, a specificity of 86%, a negative predictive value of 98% and a positive predictive value of 81% (see Table 2). 

The results from the discovery cohort analyses suggest that there is a high probability of an absent intrathecal humoral immune response in the absence of FLCκ synthesis. Therefore, we established a new laboratory workflow (Figure 2) using the FLCκ-IF as a first line analysis for the presence of an intrathecal humoral immune response. As shown in the discovery cohort, sensitivity to detect an intrathecal inflammation is reduced in patients with pathologically elevated FLCκ serum concentrations. Therefore, patient samples in which the FLCκ serum concentrations exceeded the reference range [10] (9) were excluded in an initial step in the proposed workflow (Figure 2). 

### 3.2. Part II: Evaluation of the Laboratory Workflow in a Prospective Validation Cohort

The validation cohort comprised 278 patient samples (Table 1). Of these, 32 (12%) showed an intrathecal humoral immune response as defined in the methods section. Of those, only one patient sample lacked an intrathecal FLCκ synthesis. The diagnosis of intrathecal inflammation in this patient who suffered from an epileptic seizure was based on the presence of two isolated CSF bands in isoelectric focusing. The corresponding CSF profile as well as MRI showed no signs of intrathecal inflammation. Of the prospective cohort 246 patient samples (88%) did not show an intrathecal humoral immune response as defined above. Of these, 36 had an intrathecal synthesis of FLCκ (13%). 

All together, these results yielded an excellent sensitivity of 97% and a negative predictive value of 99.5% to identify an intrathecal humoral immune response by using the FLCκ-IF as first line parameter (Figure 3, Table 2). 

If the FLCκ-IF would have been used in this cohort according to the suggested workflow, the Ig and OCB analyses would have been reduced by 208 patient samples (or 74% of all Ig and OCB analyses). Using the suggested workflow also increases the pre-test probability of a positive OCB result and thus increases the percentage of positive test results.

## 4. Discussion

While a large body of work has been reported to establish the FLCκ in CSF as a diagnostic biomarker, the final translating step from research into clinical routine is still missing. This is at least partially caused by insufficient agreement on the best method of interpretation and cut-off values [3]. As some studies suggest, the FLCκ index would be sufficient to apply FLCκ analysis in MS patients [14]. In other diseases, evaluation of FLCκ in quotient diagrams is far more accurate and less prone to false negative findings at higher or lower albumin quotient levels. Since a relevant proportion of neurological diseases are associated with an elevation of the albumin quotient [15], evaluation of the intrathecal fraction of FLCκ based on the hyperbolic reference range should be the method of choice for disease-independent evaluation [3,10,11,12,13,16]. 

Based on previous studies, we hypothesized that in the absence of intrathecal FLCκ synthesis, a humoral intrathecal immune response as defined above should also not be detectable [5,12,13,17,18]. This led us to the idea of introducing the presence of an intrathecal synthesis of FLCκ as a first line analysis for an intrathecal humoral immune response into routine CSF laboratory diagnostics. Our hypothesis is based on the pathophysiological foundation that FLCκ are only produced during the generation and assembly of Ig [17,18]. We have previously demonstrated that the FLCκ index can predict CSF specific OCB with a high sensitivity [5,12]. Furthermore, we could show that this also applies to intrathecal IgA and IgM synthesis. In contrast to Ig quotient diagrams, the intrathecal synthesis of FLCκ as determined in the quotient diagram is robust against blood contamination. This simplifies the differentiation between artificial Ig synthesis caused by blood contamination and inflammatory causes [13]. Based on this, we also decided to evaluate the patient samples with blood admixture as non-inflammatory. An elevated Q FLCκ in these samples would be an expression of a masked intrathecal humoral immune response.

Considering the results of our discovery cohort, the intrathecal fraction of FLCκ seemed suitable as first line analysis for the presence of an intrathecal humoral immune response, as only a small proportion of false negative samples were found (0.5%; NPV: 99.5%). Of these, 66% were explicable by elevated serum FLCκ levels above the serum reference range. This is in line with the concept of quotient diagrams that excessive serum concentrations of any protein of interest reduces the sensitivity to detect intrathecal synthesis of this protein (see Appendix B, Appendix A, Appendix A). Therefore, the sensitivity of the intrathecal fraction of FLCκ to detect intrathecal inflammation will be reduced in patient samples in which FLCκ serum concentrations exceed the upper reference range. We have therefore integrated the elevated serum FLCκ values into the proposed laboratory workflow used in our prospective confirmation cohort (see Figure 2). In case of elevated serum FLCκ levels, the analysis of IgG, IgA, and IgM in quotient diagrams and analysis of OCB are strongly recommended.

Only one patient sample in the discovery cohort showed an intrathecal IgM synthesis in the quotient diagram which could not be explained by an artificial blood contamination, while lacking an intrathecal fraction of FLCκ. We suspect that inacurracies in measurement of IgM due to its low concentration in CSF and its high serum gradient [19], may be the reason for this.

Appling the newly developed workflow to the prospective validation cohort resulted in an excellent sensitivity of 98% with a negative predictive value of 99.5%, confirming the findings of the discovery cohort. One patient in the validation cohort showed CSF specific borderline OCB results but no intrathecal FLCκ synthesis. Some authors suggest that two CSF bands do not reliably indicate an intrathecal IgG-synthesis [20]. 

Diagnostic laboratory workflows especially for the diagnosis of MS with a sequential determination of FLCκ and OCB have already been suggested [16,21,22]. Sanz Diaz et al. suggested the use of the FLCκ CSF concentration for a preselection, followed by the FLCκ index, which might be appropriate in case of a suspected MS [21]. However, using CSF FLCκ concentrations alone is too imprecise when applying workflows to a more unselective patient population. This does not consider the individual blood-CSF barrier function as well as the influence of serum FLCκ values, as described above. 

Our data demonstrate the benefits of including the intrathecal fraction of FLCκ as a key parameter into the laboratory workflow: In our cohorts, 62% of the Ig and OCB analysis in the discovery cohort and 74% in the validation cohort would not have been required. Thus, the implementation of this workflow would have a considerable cost-, resources- and personnel-saving effect. In addition, using the suggested workflow with a high positive predictive value for positive OCB results will avoid unnecessary OCB analysis. Certainly, if clinical findings, other laboratory parameters such as conspicuous CSF cytology, or imaging suggest an intrathecal inflammation despite a lack of intrathecal FLCκ synthesis, it will be reasonable to perform Ig quotient diagrams and OCB. Nevertheless, the use of this workflow can simplify the diagnostic pathway to exclude intrathecal humoral immune responses due to the high negative predictive value. 

The monocentric study design is a limitation of our study. On the other hand, we used a large cohort to establish the workflow. If a general diagnostic pathway is recommended, it is important that it also yields robust results for different assays. Especially OCB (e.g., silver staining vs. immunoblotting) and FLCκ determination on different platforms warrant further investigation. Noteworthy is also that the discovery cohort does not reflect the general neurological patient population due to the selectivity of the previous studies that were utilized in this cohort.

## 5. Conclusions

The presence of an intrathecal fraction of FLCκ shows an excellent sensitivity to identify an intrathecal Ig synthesis. The integration of the FLCκ-IF as a screening parameter into the routine procedures of CSF laboratories can simplify CSF laboratory evaluation and analytical effort. In addition, this will increase the pre-test probability of a positive OCB result and thus increase the percentage of positive test results.

## Figures and Tables

**Figure 1 biomolecules-12-01690-f001:**
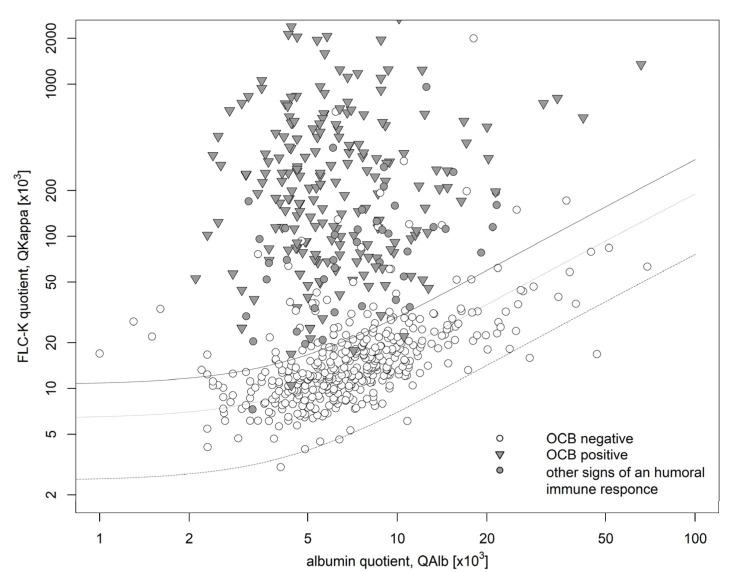
Data of the discovery cohort in a double logarithmic FLCκ quotient diagram. The bold line shows Qlim, the dashed line shows the Qmean. The bottom dashed line shows the Qlow. 302 samples (44%) showed an intrathecal FLCκ synthesis (IF FLCκ > 0) of which 60 patient samples showed no other signs of a humoral immune response (negative OCB, IF IgG/IF IgA/IF IgM < 0). 245 patient samples presented signs of an intrathecal inflammation (positive OCB (
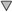
) or Ig synthesis in the corresponding quotient diagram (
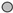
)) of which 4 lacked an intrathecal FLCκ synthesis (IF FLCκ < 0). FLCκ: free light chains kappa, Q: quotient, Alb: albumin.

**Figure 2 biomolecules-12-01690-f002:**
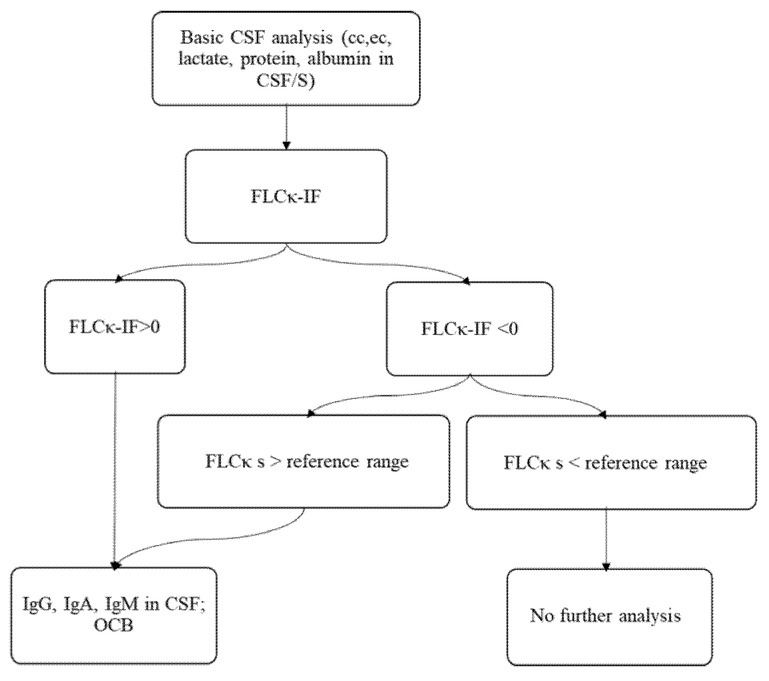
The suggested algorithm for the integration of the FLCκ-IF into the CSF laboratory workflow. After determination of cell count, erythrocyte count, lactate, and total protein, FLCκ is measured in CSF and serum and the quotient is evaluated in relation to the hyperbolic reference range. If the FLCκ-IF is >0, further analysis of immunoglobulin and OCB should be performed. If the FLCκ-IF is <0 and the serum concentration of FLCκ is normal, further analysis of Ig and OCB can be omitted. EC: erythrocyte count; OCB oligoclonal bands, Ig immunoglobulin, CSF cerebrospinal fluid, FLCκ free light chains kappa, cc cell count.

**Figure 3 biomolecules-12-01690-f003:**
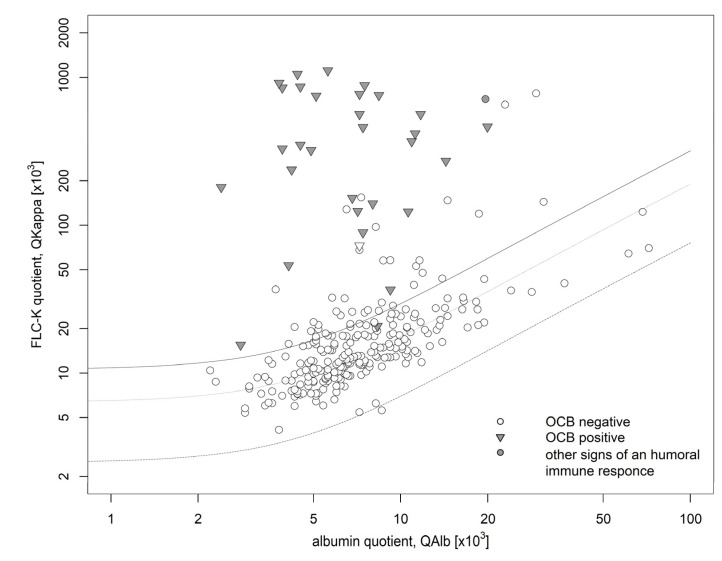
Data of the validation cohort in a double logarithmic FLCκ quotient diagram. The bold line shows Qlim, the dashed line shows the Qmean. The bottom dashed line shows the Qlow. 67 (27%) patient samples presented with an intrathecal FLCκ synthesis (IF FLCκ> 0) of which 36 patient samples showed no other signs of a humoral immune response (negative OCB, IF IgG/IF IgA/IF IgM < 0). 32 patient samples presented signs of an intrathecal inflammation (positive OCB (
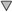
) or Ig synthesis in the corresponding quotient diagram (
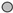
)) of which one lacked an intrathecal FLCκ synthesis (QFLCκ > QLim (FLCκ)). FLCκ free light chains kappa, Q quotient, Alb albumin.

**Table 1 biomolecules-12-01690-t001:** Continuous data are expressed as median (first; third quartiles); nominal data are given as percentages, Ig-synthesis are given as quantity and percent. CSF cerebrospinal fluid, OCB oligoclonal bands, CC cell count (leukocytes), Ig immunoglobulin, QIgG/A/M immunoglobulin G/A/M quotient, QAlb albumin quotient, FLCκ free light chains kappa.

	Discovery Cohort (*n* = 679)	Validation Cohort (*n* = 278)
Age (y)	52 (36; 65)	56 (41; 69)
QAlb	6.4 (4.7; 9.1)	6.7 (5.1; 9.4)
QIgG	3.6 (2.5; 5.1)	3.1 (2.3; 4.5)
QIgA	1.9 (1.2; 2.9)	1.7 (1.2; 2.7)
QIgM	0.4 (0.2; 0.8)	0.3 (0.2; 0.7)
One-class Ig immune response		
IgG-synthesis; *n* (%)	71 (10)	12 (4)
IgA-synthesis; *n* (%)	10 (1.4)	0
IgM-synthesis; *n* (%)	19 (2.7)	1 (0.3)
Two-class Ig immune response		
IgG/M-synthesis; *n* (%)	31 (4.5)	6 (2)
IgA/M-synthesis; *n* (%)	12 (1.8)	1 (0.3)
IgG/A-synthesis; *n* (%)	9 (1.3)	1 (0.3)
Three-class Ig immune response	4 (0.05)	2 (0.7)
Artificial blood Contamination, *n* (%)	18 (2.3)	1 (0.3)
CSF specific OCB; *n* (%)	204 (30)	31 (11)
CC/µL	1 (1; 4)	1 (1; 2)
FLCκ IF > 0%; *n* (%)	302 (44)	67 (24)
FLCκ serum (mg/L)	13.1 (10.2; 18)	13.1 (9.81; 17.97)
FLCκ CSF (mg/L)	0.363 (0.189; 1.63)	0.23 (0.12; 0.48)
QFLCκ	21.5 (12.04; 111.38)	15.21 (10.29; 26.28)

**Table 2 biomolecules-12-01690-t002:** FLCκ free light chains kappa, IF intrathecal fraction, OCB oligoclonal bands, Ig immunoglobulin.

		Evidence of Intrathecal Humoral Immune Response (OCB and/or IF IgG/A/M > 0%)	No Evidence of Intrathecal Humoral Immune Response (OCB and IF IgG/A/M < 0%)	
Discovery cohort	FLCκ-IF > 0%	244	60	PPV: 80%
	FLCκ-IF < 0%	4	374	NPV: 99%
		Sensitivity: 98 %	Specificity: 86 %	
Validation cohort	FLCκ-IF > 0%	32	36	PPV: 47%
	FLCκ-IF < 0%	1	211	NPV: 99%
		Sensitivity: 97 %	Specificity: 85 %	

## Data Availability

The data presented in this study are available on request from the corresponding author. The data are not publicly available.

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
