# Peer review of "A New Laboratory Workflow Integrating the Free Light Chains Kappa Quotient into Routine CSF Analysis"

_biomolecules, 2022, doi:10.3390/biom12111690_

Round 1

Reviewer 1 Report

This is an important study highlighting diagnostic options under a cost efficient aspect for detecting intrathecal humoral response. 

A few aspects should be revised or additionally included in the manuscript. 

  •  
  • 1. Please check numbers in the first paragraph of the results section (line 137ff) – 36% and “remaining 1%” do not add up. 

  • 2. Exclusion of relevant blood admixture was pursued for the paper and should therefore be included into the proposed algorithm. 

  • 3. OCB were not diagnosed via silver staining – this is a relevant limitation to the study and should be mentioned. 

  • 4. Please complete the cited literature for additional relevant previously conducted studies that have been published recently (i.e. doi:10.3390/brainsci12040475) 

  • 5. A multicenter approach would have been beneficial - please include this limitation into the manuscript.

Author Response

This is an important study highlighting diagnostic options under a cost-efficient aspect for detecting intrathecal humoral response. 

A few aspects should be revised or additionally included in the manuscript. 

  • The English language and style were reviewed and corrected by an English expert.
  • Please check numbers in the first paragraph of the results section (line 137ff) – 36% and “remaining 1%” do not add up.
    • The numbers have been corrected
    • 245 patient samples of 679 patients samples are ~36%
    • 4 patients samples of 245 are 1,6%. This has been changed to ~2%

  • Exclusion of relevant blood admixture was pursued for the paper and should therefore be included into the proposed algorithm.
    • Thank you very much for this comment. An intrathecal humoral immune response was defined in our work by the presence of either CSF-specific OCB and/or immunoglobulin synthesis with an IF of IgG, A, M > 0%. Blood admixture by itself, a known pitfall in CSF analysis, was not considered to indicate a humoral immune response. As OCB were performed in all samples, they would have identified intrathecal inflammation. So we did not exclude samples with blood admixture form the current analysis but considered an intrathecal IgM>IgA>IgG in the absence of OCB as artificial if erythrocytes in the CSF exceed 500/µl. We have clarified this in our manuscript.
      Furthermore as discussed on pg 8 inn row 231ff previous work of our group has demonstrated that FLCκ quotients remains negative in case of blood admixture and are therefore not prone to result in artificial intrathecal synthesis-. We therefore decided not to include it in the workflow.
  • OCB were not diagnosed via silver staining – this is a relevant limitation to the study and should be mentioned.
    • This has been added (p. 9 row 285)

  • Please complete the cited literature for additional relevant previously conducted studies that have been published recently (i.e. doi:10.3390/brainsci12040475)
    • This has been added

  • A multicenter approach would have been beneficial - please include this limitation into the manuscript.
    • This has already been stated as a limitation: p 9 row 282

Reviewer 2 Report

The Authors propose a new laboratory workflow integrating free kappa light chain (FLC kappa) intrathecal synthesis (evaluated in Reibergram) and omitting the gold standard oligoclonal IgG band (OCB) test in cases of FLC kappa IF less than 0% and normal serum FLC kappa values. The idea is not entirely new and besides references cited, similar algorithm has been proposed in the context of multiple sclerosis (MS) diagnosis by Crespi et al. already in 2017 (Crespi I et al., Combined use of kappa free light chain index and isoelectrofocusing of cerebro-spinal fluid in diagnosing multiple sclerosis: Performances and costs. Clin Lab 2017, 63: 551-9). However, Hannich et al. provide probably more accurate method for FLC kappa intrathecal synthesis estimation and more data to support this idea. The study is carefully conducted and well-written. I appreciate the explanation of false negative results of FLC kappa intrathecal synthesis in the presence of elevated serum FLC kappa concentrations in Appendix A.

The reviewer guess how the proposed algorithm will really work in daily routine practice. There are large differences between regions and countries in the policy of laboratory tests ordering, and sometimes unappropriate directive approaches of LabMed Department management towards clinicians can be encountered. What will be the Authors´ next step in this regard? Should they propose to the neurologists to order either "intrathecal Ig synthesis screening" or "intrathecal Ig synthesis complete panel"? What if FLC kappa IF be negative and other signs of intrathecal humoral immune response will be present (e.g., plasma cells in CSF cytology?) or lymphoproliferative disease with CNS infiltration will be suspected (I remember two cases of CNS lymphoma where there was isolated free lambda light chain intrathecal synthesis, although this issue is not covered in the Authors´ study.) I don´t ask the Authors to answer all these questions but would like to point out that the algorithm might be still refined in the future based predominantly on the diagnoses of these very rare cases where intrathecal humoral immune response will be missed by the FLC kappa IF test.

Minor issues:

1. p. 2 rows 54-5: the last sentence of the last paragraph could possibly be reformulated. The Authors clearly mean that FLC kappa synthesis can also be seen in the presence of IgA and IgM intrathecal synthesis, not the mere "presence" of IgA and IgM in the CSF.

2. p. 2 rows 90-96: In principle this is correct; however, e.g. a patient with neuroborreliosis where IF IgM>IF IgA>IF IgG pattern may be sometimes observed) and erythrocyte count 600/microliter could be missed by such an approach (of course, FLC kappa IF should be positive in such a patient).

3. p. 6 Figure 2 legend: eGFR (estimated glomerular filtration rate) is not mentioned anywhere in the Figure

4. p. 8 rows 260-261: physiological IgG microheterogeneity can explain serum bands (i.e., common bands in CSF and serum) but it is questionable whether CSF-restricted bands in non-inflammatory controls could be attributed to such a phenomenon

5. p. 8 rows 262 ff Consider mentioning the reference Crespi et al. Clin Lab 2017 along with refs. 15, 22 in the Discussion section as well as the reference list.

6. Appendix and Table S.1. Consider using decimal points instead of decimal commas as elsewhere in the text.

Author Response

The Authors propose a new laboratory workflow integrating free kappa light chain (FLC kappa) intrathecal synthesis (evaluated in Reibergram) and omitting the gold standard oligoclonal IgG band (OCB) test in cases of FLC kappa IF less than 0% and normal serum FLC kappa values. The idea is not entirely new and besides references cited, similar algorithm has been proposed in the context of multiple sclerosis (MS) diagnosis by Crespi et al. already in 2017 (Crespi I et al., Combined use of kappa free light chain index and isoelectrofocusing of cerebro-spinal fluid in diagnosing multiple sclerosis: Performances and costs. Clin Lab 2017, 63: 551-9). However, Hannich et al. provide probably more accurate method for FLC kappa intrathecal synthesis estimation and more data to support this idea. The study is carefully conducted and well-written. I appreciate the explanation of false negative results of FLC kappa intrathecal synthesis in the presence of elevated serum FLC kappa concentrations in Appendix A.

The reviewer guess how the proposed algorithm will really work in daily routine practice. There are large differences between regions and countries in the policy of laboratory tests ordering, and sometimes unappropriated directive approaches of LabMed Department management towards clinicians can be encountered. What will be the Authors´ next step in this regard? Should they propose to the neurologists to order either "intrathecal Ig synthesis screening" or "intrathecal Ig synthesis complete panel"? What if FLC kappa IF be negative and other signs of intrathecal humoral immune response will be present (e.g., plasma cells in CSF cytology?) or lymphoproliferative disease with CNS infiltration will be suspected (I remember two cases of CNS lymphoma where there was isolated free lambda light chain intrathecal synthesis, although this issue is not covered in the Authors´ study.) I don´t ask the Authors to answer all these questions but would like to point out that the algorithm might be still refined in the future based predominantly on the diagnoses of these very rare cases where intrathecal humoral immune response will be missed by the FLC kappa IF test.

            Thank you very much for this comment. We agree with the reviewer that such a change in laboratory will require close communication to adapt laboratory ordering. We are convinced that in the long run clinicians will appreciate rapid turn around times that can be achieved.  Of course, if clinical findings, other laboratory parameters such as conspicuous CSF cytology, or imaging suggest intrathecal inflammation despite the absence of intrathecal FLCκ synthesis, it will be reasonable to perform Ig quotient diagrams and OCB. This was added to the text (p. 9, row 278-280)

Minor issues:

  1. p. 2 rows 54-5: the last sentence of the last paragraph could possibly be reformulated. The Authors clearly mean that FLC kappa synthesis can also be seen in the presence of IgA and IgM intrathecal synthesis, not the mere "presence" of IgA and IgM in the CSF.

            This has been reformulated

  1. p. 2 rows 90-96: In principle this is correct; however, e.g. a patient with neuroborreliosis where IF IgM>IF IgA>IF IgG pattern may be sometimes observed) and erythrocyte count 600/microliter could be missed by such an approach (of course, FLC kappa IF should be positive in such a patient).

            We agree with the reviewer that this is an important caveat in current CSF analysis that would be detected by inclusion of FLC kappa IF in the analysis. However, OCB can be expected to be positive in such a case of neuroborreliosis marking the sample as inflammatory in our current study.  Similarly, an elevated QFLCk in these samples would be an expression of a masked intrathecal humoral immune response. This has been added to the text p8 row 235-237.

  1. p. 6 Figure 2 legend: eGFR (estimated glomerular filtration rate) is not mentioned anywhere in the Figure

            This was removed

  1. p. 8 rows 260-261: physiological IgG microheterogeneity can explain serum bands (i.e., common bands in CSF and serum) but it is questionable whether CSF-restricted bands in non-inflammatory controls could be attributed to such a phenomenon

            This was removed

  1. p. 8 rows 262 ff Consider mentioning the reference Crespi et al. Clin Lab 2017 along with refs. 15, 22 in the Discussion section as well as the reference list.

            This was added

  1. Appendix and Table S.1. Consider using decimal points instead of decimal commas as elsewhere in the text.

            Decimal commas were changed to decimal points.
